# Molecular Characterization and Expression Profile of *PaCOL1*, a *CONSTANS*-like Gene in *Phalaenopsis* Orchid

**DOI:** 10.3390/plants9010068

**Published:** 2020-01-04

**Authors:** Yi-Ting Ke, Kung-Fu Lin, Chu-Han Gu, Ching-Hui Yeh

**Affiliations:** Department of Life Sciences, National Central University, Taoyuan 32001, Taiwan; kortinn@yahoo.com.tw (Y.-T.K.); gabriel.kfl@gmail.com (K.-F.L.); chuhangu@gmail.com (C.-H.G.)

**Keywords:** BiFC, biological clocks, circadian rhythms, *CONSTANS*, floral transition, *Phalaenopsis aphrodite*, photoperiodism, Y2H

## Abstract

*CONSTANS* (CO) and *CONSTANS*-like (COL) genes play important roles in coalescing signals from photoperiod and temperature pathways. However, the mechanism of CO and COLs involved in regulating the developmental stage transition and photoperiod/temperature senescing remains unclear. In this study, we identified a *COL* ortholog gene from the Taiwan native orchid *Phalaenopsis aphrodite*. The *Phalaenopsis aphrodite* CONSTANS-like 1 (PaCOL1) belongs to the B-box protein family and functions in the nucleus and cytosol. Expression profile analysis of *Phalaenopsis aphrodite* revealed that *PaCOL1* was significantly expressed in leaves, but its accumulation was repressed during environmental temperature shifts. We found a differential profile for PaCOL1 accumulation, with peak accumulation at late afternoon and at the middle of the night. *Arabidopsis* with *PaCOL1* overexpression showed earlier flowering under short-day (SD) conditions (8 h/23 °C light and 16 h/23 °C dark) but similar flowering time under long-day (LD) conditions (16 h/23 °C light and 8 h/23 °C dark). Transcriptome sequencing revealed several genes upregulated in *PaCOL1*-overexpressing *Arabidopsis* plants that were previously involved in flowering regulation of the photoperiod pathway. Yeast two-hybrid (Y2H) analysis and bimolecular fluorescence complementation (BiFC) analysis revealed that PaCOL1 could interact with a crucial clock-associated regulator, AtCCA1, and a flowering repressor, AtFLC. Furthermore, expressing *PaCOL1* in *cca1.lhy* partially reversed the mutant flowering time under photoperiod treatment, which confirms the role of PaCOL1 function in the rhythmic associated factors for modulating flowering.

## 1. Introduction

Throughout their life span, plants are constantly exposed to changing environmental conditions. The sessile plants have to develop multitudinous strategies to survive the environmental challenges. The intimations of transition from vegetative to reproductive growth are determinant in the life cycle of plants. To ensure the inception of flowering at the correct time, plants have to coalesce a diverse range of external and internal signals [1]. The photoperiod oscillator, ambient temperature, gibberellic acid, vernalization, and autonomous/aging pathways are the main factors involved in the transition of plants from the vegetative to reproductive stage [2,3,4,5].

The use of genetic approaches and functional analyses has identified several key moderators in the complex signal regulatory network in *Arabidopsis thaliana*; most of these moderators belong to the regulatory protein family [6,7]. Among the regulatory protein families, B-box (BBX) proteins are a group of transcription factors with one or more B-box domains that can be stabilized by binding to zinc ions [8]. The CONSTANS (CO), CONSTANS-like (COL), TIMING of CAB1 (TOC) motif is a common conserved domain in BBX proteins [9]. These proteins are involved in protein–protein modulation and gene regulation [10,11,12,13,14].

Overall, 32 *Arabidopsis* BBX proteins [15], 30 rice BBX proteins [16], 29 tomato BBX proteins [17], and 25 pear BBX proteins [18] have been identified, with diverse functions. According to the structural characteristics of their conserved domains and phylogenetic comparisons, plant BBX proteins may be classified into five groups: groups 1 and 2 have two tandem B-boxes plus a CCT motif; group 3 has a B-box plus a CCT motif; group 4 has two B-boxes; and group 5 has only one B-box [19,20]. In *Arabidopsis*, the BBX proteins are well known for their functions in regulating plant growth, seed germination, seedling photo-morphogenesis, and shade avoidance, acting as antistressors to biotic/abiotic stresses and sensing the light as well as circadian signaling changes [20,21]. CONSTANS (CO; also named AtBBX1), the best-characterized BBX protein, is a master integrator for the flowering regulation pathway by mediating the input signals of temperature and light to control flowering time in *A. thaliana* under long-day (LD) conditions [15,22]. AtBBX1 associates with the nuclear factor Y subunit B (NF-YB)/nuclear factor Y subunit C (NF-YC) dimer to form a trimer complex via a CCT motif-dependent interaction, which then binds the promoter of *FLOWERING LOCUS T* (*FT*) to trigger gene expression and flowering [23]. In contrast, a rice *CO* homolog, *heading date 1* (*Hd1*/*OsBBX18*), represses flowering under LD conditions but accelerates it under short-day (SD) conditions by modulating the expression of *Hd3a*, a rice *FT* homolog [24,25,26]. Further studies indicated that phosphorylation by a rice *casein kinase II α subunit* (*CK2α*), *Hd6*, modulates the LD floral suppression of the *Hd1*-dependent pathway [27]. However, *Arabidopsis CK2α* subunit triple mutants showed reduced FT expression and delayed flowering under LD conditions [28]. Thus, some basic components of the genetic network modulating flowering are conserved in *Arabidopsis* and rice, but their functions have diverged and show completely different flowering responses [29].

CONSTANS-like (COL) proteins also have divergent functions in flowering regulation in *Arabidopsis* and other plants [21,30]. COL proteins form a homomultimer or associate with other proteins including other BBXs to modulate flowering time with distinct mechanisms [31,32]. AtCOL3/AtBBX4-AtBBX32 dimers control flowering time via binding the promoter of the florigen *FT* gene [33,34]. However, COLs can function as a negative regulator in controlling flowering time. Overexpression of *AtCOL9*/*BBX7* in *Arabidopsis* could delay flowering time under LD conditions by providing a feedback effect on CO and FT expression [30]. A similar downregulation of rice florigen genes *Hd3a* and *RFT1* was identified in late-flowering *OsCOL9* and *OsCOL10* transgenic rice plants [21,35]. COLs play significant roles in photoperiodic flowering in many plants, but their functions remain unknown in non-model systems.

The *Orchidaceae*, one of the two largest families of flowering plants, have about 28,000 species with flowers that are often colorful and scented. *Phalaenopsis* is a monopodial orchid with a high market value. *Phalaenopsis aphrodite* subsp. *formosana* (*P. aphrodite*), one of the white flower species of moth orchid native to Taiwan, is usually used to investigate the mechanism for flowering regulation and floral organ development of *Phalaenopsis* [36,37]. Recent studies indicated that *P. aphrodite FT1* (*PaFT1*), an *Arabidopsis FT* ortholog, interacted with *PaFD*, a bZIP transcription factor and abundant in developing inflorescence, to promote flowering by integrating signals from ambient temperatures suitable for floral induction [38,39]. Gain- and loss-of-function analyses indicated that *P. aphrodite LEAFY* (*PaLFY*), an *Arabidopsis LFY* ortholog, might function in heading time or floral organ identity gene expression in *Arabidopsis*, rice, and orchids [40]. In Oncidium orchid, a perennial epiphytic plant with importance for the floriculture industry in Taiwan, high temperatures induced *cytosolic ascorbate peroxidase 1* (*OgcytAPX1*) expression to reduce H_2_O_2_ accumulation and ascorbate redox ratio, which caused the transition from the vegetative to the reproductive stage [41]. However, the floral inductive pathways at the genetics level are still unclear.

We used rapid amplification of cDNA ends-PCR (RACE-PCR) and genomic walking strategies with CCT motif-specific degenerated primers to isolate a *COL* gene, *PaCOL1*, from floral buds of *P. aphrodite* (mRNA/CDS GeneBank accession no: FJ810423/ACY95395.1; promoter GeneBank accession no: HM068557; Appendix A). PaCOL1 was the first member of the BBX family identified in the native *P. aphrodite* (diploid). Biochemical and molecular analyses revealed that PaCOL1, a nuclear localized protein, could interact with specific transcription factors involved in sensing day length and temperature-alteration signals. Ectopic overexpression analyses revealed that *PaCOL1* could function in mediating flowering time in *Arabidopsis* plants.

## 2. Results

### 2.1. PaCOL1 Encodes a COL Protein

The genomic DNA (gDNA) of *PaCOL1* is 945 bp, which splices into a 744-bp sense RNA product and encodes a protein with 247 amino acids of 27.9 kDa (Appendix A). From the deduced amino acid sequence, we identified one B-box zinc finger domain near the N-terminal region and one CCT motif at the C-terminal region (Appendix A). Also, the CCT motif included a putative nuclear localization sequence (NLS). Multiple sequence alignment was carried out with the Clustal Omega program with default parameters in the European Bioinformatics Institute toolbox (https://www.ebi.ac.uk/Tools/msa/clustalo/) (Figure 1A). The results revealed PaCOL1 with 69–78% and 60–81% similarity to *Arabidopsis* COL proteins and other plant COL proteins, respectively (Figure 1B). To investigate the evolutionary relationship among plant COLs, we constructed a phylogenetic tree by using the Mega6 software (http://megasoftware.net/) and unweighted pair-group method with arithmetic means (UPGMA) method. Phylogenetic tree analysis with other COL/BBX proteins from *Arabidopsis*, rape, sesame, rice, corn, grass, barley, darnel, and orchid revealed the sequence association as three major clades in the evolution (Figure 1C). The members of the PaCOL1 clade share a B-box domain and a CCT motif, which suggests that PaCOL1 shared the same sequence structures as the group 3 B-box protein family in *Arabidopsis* and rice.

### 2.2. Subcellular Analysis of PaCOL1 Protein

To determine the subcellular localization of PaCOL1, the corresponding coding sequence was genetically fused with green fluorescence protein (GFP) in transient expression vectors. Recombinant constructs were expressed in onion epidermal cells by using particle bombardment and transient expression assay. The vector containing *eIF4AIII* [42] was used as a marker to indicate the nucleus region or even nuclear body. The full length of PaCOL1 protein localized in the nucleus (Figure 2). In addition, we analyzed the cellular localization of PaCOL1 protein without a B-box domain (deletion of amino acid residues 6–52; PaCOL1ΔB) or CCT motif (deletion of amino acid residues 177–221; PaCOL1ΔC). PaCOL1ΔB was still localized in the nucleus, but PaCOL1ΔC localized in the cytoplasm and nucleus. We further confirmed the cellular localization of PaCOL1, PaCOL1ΔB, and PaCOL1ΔC in protoplasts from *P. aphrodite* seedling root cells (Appendix A). Thus, PaCOL1 may function in the nucleus.

### 2.3. Expression Profiles of PaCOL1

To investigate the function of *PaCOL1* in *P. aphrodite*, we tested expression profiles in vegetative and reproductive tissues. Under normal growth conditions (12 h/23 °C light and 12 h/23 °C dark, 58.39 W/m^2^), *PaCOL1* transcripts were significantly expressed in buds, floral stems, and leaves but were less accumulated in flowers and roots (Figure 3A). Leaves are important for sensing light and temperatures. In *Arabidopsis*, AtBBX1/CO was synthesized in leaf cells, followed induced florigen production, and translocated to shoot apical for floral induction [14,22]. The orchid market often uses low ambient temperatures (i.e., 12 h/23 °C light and 12 h/18 °C dark for four weeks) for floral bud induction [43]. Thus, we examined the expression levels of *PaCOL1* of mature *P. aphrodite* plants with floral bud induction or high temperature treatment (12 h/32 °C light and 12 h/23 °C dark). We found that *PaCOL1* transcript level in leaves was significantly decreased after floral bud induction (Figure 3BII) but recovered to a high level when the plants returned to normal growth conditions (Figure 3BIII). In contrast, *PaCOL1* level in leaves did not increase when plants were transferred to high temperature treatment followed by floral bud induction treatment (Figure 3BIV). *PaCOL1* transcript levels did not differ in roots and stems of *P. aphrodite* plants during treatments.

The expression profiles of *COL* genes from many plants are often differentially regulated by light and exhibit a diurnal rhythm [44,45,46]. Next, we investigated the daily expression pattern of PaCOL1 protein. On immunostaining with PaCOL1 antibody (Appendix A), PaCOL1 protein (~28 kDa) was clearly detected from late afternoon to the middle of the night (Figure 3C). These results revealed that PaCOL1 protein differentially accumulated in *P. aphrodite* under 12 h light/12 h dark conditions.

### 2.4. Overexpression of PaCOL1 in Arabidopsis Conferred SD Floral Inductivity

To further investigate the roles of *PaCOL1*, we developed transgenic *A. thaliana* overexpressing *PaCOL1* in ecotype *Col-0* plants. We used T3 homozygous transgenic lines COL1-ox, COL1ΔB-ox, and COL1ΔC-ox constitutively expressing *PaCOL1*, *PaCOL1ΔB*, and *PaCOL1ΔC*, respectively, for further study. We measured and compared plant height, leaf morphology, and flowering time between wild type (WT) and transgenic plants. Flowering time was calculated as the number of rosette leaves when the bolt was at 1 cm length. Under LD conditions (16 h/23 °C light and 8 h/23 °C dark), plant height and leaf size did not differ among WT, COL1-ox, COL1ΔB-ox, and COL1ΔC-ox plants. In contrast, COL1ΔB-ox plants flowered significantly later than the WT, with no difference in flowering time between COL1-ox as well as COL1ΔC-ox plants and the WT (Figure 4A). Under SD conditions (8 h/23 °C light and 16 h/23 °C dark), COL1-ox plants had smaller size rosette leaves, necrosis symptoms on rosette leaves, and more branches on the major floral stem as compared with the WT (Appendix A). In addition, COL1-ox and COL1ΔB-ox plants showed an early-flowering phenotype (18% and 24% decrease in number of rosette leaves, respectively) as compared with WT plants, whereas the flowering time of COL1ΔC-ox plants was similar to that of the WT (Figure 4A and Appendix A).

To further evaluate the abundance of target mRNAs mediated by PaCOL1 and unravel the molecular mechanism regulated by this gene, we analyzed transcriptome of shoot apex from 40-day-old SD-grown transgenic plants with the Illumina HiSeq 4000 platform (Novogene, Hong Kong). After filtering, we obtained more than 42,000,000 clean reads (Figure 4B). Relative fold change was calculated by comparing the RNA sequencing (RNAseq) data for COL1-ox1 and WT plants grown under SD conditions, with relative change of ≥2-fold considered significantly different. We identified 1370 upregulated and 1269 downregulated genes in COL1-ox1 versus WT plants (Appendix A). On Gene Ontology (GO) analysis (http://geneontology.org/), the genes upregulated in COL1-ox1 were classified into 22 groups (Figure 4C). For annotated genes (n = 234) involved in the developmental process, we found 48 (3% in upregulated dataset) corresponding to the flower-development–associated process (Figure 4D). In addition, we found 78 upregulated genes including MYBs, MADS-box proteins, bZIPs, ATPases, helicases, ubiquitination enzymes (UEs), and zinc-finger proteins as putative flowering-regulation candidate genes [47,48,49,50]. Thus, PaCOL1 may affect transcript levels of flowering-related *Arabidopsis* genes under SD conditions.

Furthermore, we used qRT-PCR to investigate the expression of *AtCCA1* (At2G46830), *AtCO*, *AtSVP* (At2G22540), *AtSOC1* (At2G45660), *AtFLC* (At5G10140), *AtAGL24* (At4G24540), and *AtTSF* (At4g20370), which are important floral modulator genes but were not found in RNAseq data. Under SD conditions, the expression of *AtCO* was higher in 40-day-old COL1-ox1, COL1ΔB-ox1, and COL1ΔC-ox1 plants than in the WT, whereas the expression of *AtCCA1*, *AtFLC*, *AtSOC1*, *AtSVP*, and *AtAGL24* genes was nearly equal in WT and transgenic plants (Figure 4E). In addition, we found the expression of *CDF1* (At5G62430), a repressor of *CO*, was decreased in COL1-ox1 transgenic plants but the accumulation of *MAF5* (At5G65080), an *FLC*-like gene, was slightly increased in COL1-ox1 transgenic plants. A similar expression level of *TFL1* (At5G03840), an antagonist of FT, was found in WT and transgenic plants. We hardly detected the accumulation levels of *AtTSF* transcripts in this assay (data not shown). We also found no difference in accumulation of *NFYB2* (At5g47640) and *NFYC3* (At1g54830) transcripts, the positive regulators for flowering under LD conditions. Of note, *GI* (At1G22770), involved in regulation of circadian rhythm and photoperiodic flowering, was shown to upregulate in the RNAseq dataset but to downregulate in qRT-PCR analysis. Our results suggest that ectopic expression of *PaCOL1* in *Arabidopsis* may disturb the circadian output gene expressions.

### 2.5. PaCOL1 Can Interact with AtCCA1 and AtFLC

The interaction between PaCOL1 and other proteins was predicted by using the STRING v 11 online program [51]. The results suggest a possible link between PaCOL1 and AtCCA1/AtLFY, BBXs, or transcriptional factors in clock control in *Arabidopsis* (Figure 5A). We thus used yeast two-hybrid (Y2H) analysis to examine the protein–protein physiological interaction between PaCOL1 and AtSOC1, AtSVP, AtFLC, and AtCCA1. Hybrid yeast cells containing PaCOL1 with AtCCA1 and AtFLC could survive on triple amino acid dropout selection medium (Figure 5B and Appendix A). In addition, two truncated PaCOL1 proteins, PaCOL1ΔB and PaCOL1ΔC, could interact with AtFLC and AtCCA1, which was confirmed by using the quanta dropout medium (Figure 5C). We did not find an interaction between PaCOL1 and AtSOC1 or AtSVP (Figure 5B).

Furthermore, we used bimolecular fluorescence complementation (BiFC) analysis to examine the PaCOL1–AtCCA1 and PaCOL1–AtFLC interactions in onion epidermal cells and orchid protoplast cells. PaCOL1, PaCOL1ΔB, and PaCOL1ΔC interacted with AtFLC (Figure 5D). In addition, we detected AtCCA1 interacting with PaCOL1, PaCOL1ΔB, and PaCOL1ΔC (Figure 5E). We observed the yellow fluorescence protein (YFP) fluorescent signal for all combinations tested in the nucleus. We did not detect any YFP fluorescence in the combination of PaCOL1 with AtSVP and AtSOC1 (data not shown). No interactions of PaCOL1 or AtFLC with vector only were observed (Figure 5F). In addition, for co-transfection of the PaCOL1 full length fused with each half of the YFP gene, we observed the YFP florescent signal in orchid protoplasts, but there were no signals in YFP^C^-PaCOL1∆B/YFP^N^-PaCOL1∆B, YFP^C^-PaCOL1∆C/YFP^N^-PaCOL1∆C, or YFP^C^/PaCOL1-YFP^N^ (Appendix A), which suggests that PaCOL1 protein may form dimers for their physiological functions. Thus our Y2H and BiFC results suggest specific domain(s) other than the B-box domain and CCT motif for the interaction of PaCOL1 with AtCCA1 and AtFLC.

### 2.6. PaCOL1 Partially Complemented cca1.lhy Mutant Flowering Time under SD Conditions

AtCCA1 and its paralog late elongated hypocotyl (LHY) are important oscillators in *Arabidopsis* [52]. We obtained a cca1.lhy double mutant (CS9812), showing photoperiod-insensitive early flowering under LD and SD conditions and lost rhythms in clock-controlled gene transcription under constant light [53,54], from The Arabidopsis Information Resource (TAIR) (https://www.arabidopsis.org/). We used the mutant to investigate whether PaCOL1 and AtCCA1 have functional redundancy because of their interaction. COL1-ox1, COL1-ox2, COL1ΔB-ox1, COL1ΔB-ox2, COL1ΔC-ox1, and COL1ΔC-ox2 plants were individually crossed with cca1.lhy double mutant. Lines *cca1.lhy*/COL1-ox1, *cca1.lhy*/COL1-ox2, *cca1.lhy*/COL1ΔB-ox1, *cca1.lhy*/COL1ΔB-ox2, *cca1.lhy*/COL1ΔC-ox1, and cca1.lhy/COL1ΔC-ox2 were obtained for further analysis. Among the double mutants, *cca1.lhy*/COL1ΔC-ox1 plants exhibited increased number of axillary branches but had similar rosette leaf numbers as *cca1.lhy*/COL1ΔC-ox2 plants under LD and SD conditions. Thus we calculated the number of rosette leaves for analyzing flowering time of the six double mutants under LD and SD treatments. As compared with *cca1.lhy* plants, *cca1.lhy*/COL1-ox and *cca1.lhy*/COL1ΔB-ox plants flowered significantly later under LD and SD conditions (Figure 6). In contrast, the flowering time of *cca1.lhy*/COL1ΔC-ox and cca1.lhy plants was similar under LD conditions, and *cca1.lhy*/COL1ΔC-ox1 showed a late-flowering phenotype under SD conditions. PaCOL1 may partially compensate for the loss of CCA1 and LHY function during the regulation of flowering time. Using crossing methods, the PaCOL1 gene was also introduced into three early flowering mutants, *bbx5* (SALK_149180C, *Col-0* background; group-1 BBX), *bbx6* (SALK_096361, *Col-0* background; group-1 BBX), and *bbx8* (SALK_061961C, *Col-0* background; group-2 BBX), and a late flowering mutant ft (CS185, L*er-0* background) under LD. However, there was no significant difference in flowering time between progenies and parental lines (Appendix A).

## 3. Discussion

Flowering is a seasonal response that involves distinct signaling pathways. BBX proteins have positive or negative effects on the complex regulation of flowering [31,32,33]. Under SD conditions, overexpression of *OsBBX27/OsCO3* in rice had opposing roles in the expression of FT homologs (Hd3a and FT-like) and resulted in a late-flowering phenotype [55]. Expression of *SiCOL1*, *SiCOL2*, *CO* homologs of sesame, enhanced *FT* expression in transgenic *Arabidopsis* but could repress the expression of *FT* homologous genes in sesame under LD conditions [56]. With knowledge of the conserved sequence of the B-box domain and CCT motif, we screened an orchid COL protein from *P. aphrodite*, PaCOL1, which contains a B-box domain and a CCT motif (Figure 1A). Recently, *PaCOL1* was reported in the Orchidstra database open source (http://orchidstra2.abrc.sinica.edu.tw/orchidstra2/index.php) as the *CONSTANS-like 1* gene (accession no. PATC130360) [57]. Phylogenetic tree analysis of BBXs indicated that PaCOL1 and some group-3 COL proteins of rice and *Arabidopsis* are in the same clade (Figure 1C), but have little physiological research for group-3 BBX proteins. Our subcellular analysis agreed that the CCT motif is important for PaCOL1 localization to the nucleus (Figure 2 and Appendix A). CCT-initiated nuclear localization has been found necessary for the BBX24 protein function in light signaling [58]. From our research, PaCOL1 may function as an integrator for flowering by modulating the input signals of the diurnal rhythmic and stress-induced pathways in *P. aphrodite*. Also, our results suggest that PaCOL1 may be involved in the plant development (Figure 6). However, these require further investigation.

The circadian clock is an approximately 24 h biological oscillator, which generally enables organisms to adjust their physiological activities corresponding to the external light/dark cycles by expecting daily environmental changes. Expression profile analysis of *PaCOL1* revealed suppressed *PaCOL1* accumulation in leaves during the temperature shift (Figure 3B). On immunoblotting analysis, PaCOL1 protein accumulation showed a differential profile, with a peak during late afternoon and at the middle of the night in *P. aphrodite* (Figure 3C).

Vernalization is a key role for many plants to adapt to a temperature cue for flowering and germinating under suitable conditions [59,60]. VRN2 and its orthologs, with high conserved sequences during evolution, are often repressors of flowering in monocot cereals under LD conditions [61]. In this study, continuous expression of *PaCOL1* in *Arabidopsis* plants caused early flowering under noninductive SD conditions. However, WT and transgenic plants with a long period of cold temperature during seed imbibition (Appendix A) or seedling stage (Appendix A) showed similar flowering time under SD conditions. Therefore, PaCOL1 may not participate directly in the cold-induced flowering regulatory pathway. From our RNAseq results, we found that the expression of *PaCOL1* could upregulate (≥2-fold increase) genes for the flowering regulatory network in *Arabidopsis* under SD conditions (Appendix A). Among these genes, ATPases, helicases, DEAD-box proteins, WDRs, SETs, and PHD proteins are involved in chromatin-associated transcriptional regulation [62,63,64]; MADS-box proteins, MYBs, AP2s, C2H2 zinc finger proteins, bZIPs, ABI3s, bHLHs, UEs, hydrolases, and IQ domain proteins (IQDs) are important for the transition from the vegetative to reproductive stage [38,48,65,66]. Expression of an *IQD* gene in *Arabidopsis* may stimulate floral transition by activating *AtCO* [67]. Our real-time PCR results revealed an increased *AtCO* transcript level in transgenic *Arabidopsis* overexpressing *PaCOL1* variants under SD conditions (Figure 4E). The expression of circadian clock output gene *GI* or *CDF* also interfered. Thus, PaCOL1 may function in the photoperiodic flowering control pathway in *Arabidopsis*.

The circadian network is conserved in higher plants [68]. Plants adapt to diurnal changes with regulation via a number of negative and positive feedback loops. CCA1, a key clock-associated modulator in *Arabidopsis*, functions with LHY in a negative feedback loop. Light transiently induces *CCA1* and *LHY* transcription, with the peak at dawn [69,70]. CCA1 and LHY proteins, MYB transcription factors, repress the expression of the evening-expressed gene *Timing of Cab expression 1* (*TOC1*) by directly binding to the *TOC1* promoter. Synergistic functioning of CCA1 and LHY to TOC1 forms a negative feedback loop in the plant circadian regulation network [71,72]. In the early morning, CCA1 and LHY transiently increased *Pseudo-response regulator 7* (*PRR7*) and *PRR9* expression [73]; however, the PRR5, PRR7, and PRR9 protein complex repressed *CCA1* and *LHY* transcription, which resulted in another negative feedback loop to each other [74,75].

Using Y2H and BiFC analysis, we identified that PaCOL1 and its truncated forms were able to interact with AtCCA1 and AtFLC in the nucleus (Figure 5). In addition, our complementary test revealed that expression of *PaCOL1* in the *cca1.lhy* mutant partially rescued the flowering time of the mutants under LD and SD conditions (Figure 6). However, PaCOL1, CCA1, and LHY show much divergence in amino acid sequence. Using the NetPhos 3.1 server (http://www.cbs.dtu.dk/services/NetPhos/) to predict the eukaryotic protein phosphorylation sites, we identified a PaCOL1 16-amino acid region with a serine phosphorylation site showing similarity with a conserved domain for phosphorylation activity of AtCCA1 [76]. Also, we found similar regions in AtBBX1, OsHd1, and HvCO1 (Appendix A). Phosphorylation of the conserved amino acids in CCA1 and LHY was necessary for their stability to control flowering time [27,76]. However, we need further study to identify the role of the 16-amino acid region. In addition, AtCCA1 is a master regulator of reactive oxygen species (ROS) homeostasis [77], so PaCOL1 may also be involved in the ROS response induced by light length or temperature shifting.

Floral transition is an important stage for plants to overcome environmental challenges to survive. *FT*-related genes are crucial for accelerated flowering under LD conditions [78,79] and are important to accelerate flowering in some plants under SD or stress conditions [38,47,80]. Thus, PaCOL1 may play a key role in the photo-rhythmical flowering pathways in *P. aphrodite*. Of note, deletion of the CCT motif or B-box domain did not lose the interaction between PaCOL1 and AtCCA1 and AtFLC (Figure 5), whereas expression of *PaCOL1ΔC* could not affect the flowering time of *Arabidopsis* plants under SD conditions (Figure 4A and Figure 6). Hence, PaCOL1 might interact with these two flowering modulators via other specific domain(s) although CCT motif and B-box domain are well known for protein–protein interaction in BBX proteins. Identification of the specific region needs further study.

## 4. Materials and Methods

### 4.1. Plant Materials

*P. aphrodite* (diploid) was grown in the culture room under a 12 h light/12 h dark (23 ± 1 °C) cycle. The plants at five-leaf stage (mature stage) were used for bolting induction. The bolting induction plants were cultured in an incubator under 23 °C/18 °C (day/night) oscillation with about 100 μmolm^−2^s^−1^ light for four weeks. After bolting induction, the plants were grown in a 12 h light/12 h dark (23 ± 1 °C) culture room. The leaves for PCR amplification/protein extraction were collected from independent plants with at least three biological replicates for each sampling and analysis.

*A. thaliana* ecotype *Col-0* was used as the wild type (WT). The *cca1.lhy* mutant in the Wassilewskija (Ws) background was obtained from The Arabidopsis Information Resource (TAIR) (https://www.arabidopsis.org/). *Arabidopsis* plants were grown at 23 °C in a 16 h light/8- h dark (LD) or 8 h light/16 h dark (SD) cycle in a growth chamber with 100 μmolm^−2^s^−1^ light and 60% relative humidity. *Arabidopsis* seeds were sterilized with 1.2% bleach and sown on half-strength Murashige and Skoog (1/2 MS) medium containing 1% sucrose (w/v), 0.8% agar (w/v), buffered to pH 5.7 with 25 µg/mL hygromycin for homozygous transgenic line selection. Flowering time was determined as the number of rosette leaves when the bolt was at 1 cm length.

### 4.2. RT-PCR and qRT-PCR

Transcript levels in different tissues were analyzed by RT-PCR. Total RNA was extracted as described [81]. The first-strand cDNA was synthesized from 2 μg total RNA in a 20 μL reaction volume with use of the SuperScriptIII First-Strand Synthesis system(Invitrogen, Thermo Scientific, Waltham, MA, USA). RNA and cDNA were quantified by using a spectrophotometer (NanoDrop 1000, Thermo Scientific, Waltham, MA, USA). PCR amplification conditions were 27 cycles of 30 s at 95 °C, 30 s at 55 °C, and 50 s at 68 °C, then 5 min at 68 °C. Primers used for analyzing gene expression levels were designed with the use of Primer3 (http://bioinfo.ut.ee/primer3-0.4.0/) and are shown in Appendix A. DNA from 15 μL of each PCR reaction underwent 1.2% (w/v) agarose gel electrophoresis. ImageJ (http://rsbweb.nih.gov/ij/) was used to quantify the ethidium-bromide-stained DNA bands. The expression of *Phalaenopsis* actin9 (PaACT9) and *Arabidopsis β-tubulin* (AtTUB) was the internal control.

qRT-PCR involved using the iCycler-iQ5 Multicolor Real-Time PCR Detection System (Bio-Rad, Hercules, CA, USA) and KAPA SYBR FAST qPCR Master Mix (2x) Kit (KAPA Biosystem, Wilmington, MA, U.S.A.) with the primer sequences listed in Appendix A. qRT-PCR reaction involved 20 ng cDNA with each set of primers and the KAPA SYBR FAST qPCR buffer Master Mix. PCR cycling included an initial step at 95 °C for 5 min, then 40 cycles of 10 s at 95 °C, 30 s at 55 °C, and 20 s at 68 °C. The comparative C_T_ method was used to measure the relative amount of each sample. The expression of PaACT9 was the internal control.

### 4.3. Preparation of DNA Constructs and Transformation

The DNA fragment of *CaMV35S*-*Nos* with cloning sites for *Xba*I/*BamH*I/*Kpn*I was cut from *pBI121* shuttle vector and subcloned into *pCAMBIA1301*. Then, the *Xba*I/*BamH*I fragments of PaCOL1, PaCOL1ΔB, and PaCOL1ΔC were separately cloned into the vector in the sense orientation downstream of the *CaMV35S* promoter to generate the overexpression vectors pCOL1-ox, pCOL1ΔB-ox, and pCOL1ΔC-ox. Sequences for all primers are in Appendix A. All constructs were verified by sequencing analysis. The recombinant vectors were transferred to *Agrobacteria* (*GV3101*) and used to transform *Arabidopsis* by the floral-dip technique [82]. Transformed *Arabidopsis* seedlings were selected as described [81]. Homozygous T2 transgenic plants were selected by growth of the T3 generations on 1/2 MS medium containing 50 mg/l hygromycin, 1% sucrose (w/v), and 0.8% agar. T3 transgenic seeds derived from homozygous T2 transgenic plants were used for subsequent tests.

Transgenic plants COL1-ox1, COL1-ox2, COL1ΔB-ox1, COL1ΔB-ox2, COL1ΔC-ox1, and COL1ΔC-ox2 were selected to cross with the *cca1.lhy* mutant. F2 lines *cca1.lhy*/COL1-ox1, *cca1.lhy*/COL1-ox2, *cca1.lhy*/COL1ΔB-ox1, *cca1.lhy*/COL1ΔB-ox2, *cca1.lhy*/COL1ΔC-ox1, and *cca1.lhy*/COL1ΔC-ox2 were obtained to produce the F3 generation, which was used for analyzing flowering time.

### 4.4. Subcellular Localization Assays

Fragments of PaCOL1, PaCOL1∆B, and PaCOL1∆C were PCR-amplified with the primers listed in Appendix A. PCR products were individually fused upstream of the GFP at the *Xba*I/*BamH*I sites in the *p326GFP* vector (*35S::GFP*) [83], which resulted in constructs under control of the *CaMV35S* promoter. The plasmids were transiently expressed in onion epidermal cells by particle bombardment as described [81] and further confirmed in orchid protoplast cells by PEG transfection [84]. A helium biolistic particle-delivery system (model PDS-1000, Bio-Rad, Hercules, CA, USA) was used for particle bombardment. In total, 5 µg recombinant plasmid DNA was used per transformation, and all target materials were bombarded twice. The bombarded onion epidermal cells were recovered at 25 °C in the dark for at least four hours. The samples were observed by using an Olympus U-LH 100 HG inverted fluorescence microscope.

### 4.5. Yeast Two-Hybrid (Y2H) Analysis

For Y2H assay, the bait DNA fragments PaCOL1, PaCOL1∆B, and PaCOL1∆C were PCR-amplified and cloned into the pGBKT7 expression vector (Clontech, Mountain View, CA, USA). The prey DNA fragments AtFLC, AtSVP, AtSOC1, AtCCA1, PaAPX, and PaFT were PCR-amplified and cloned into the *pGADT7* expression vector (Clontech, Mountain View, CA, USA). All constructs were verified by sequencing analysis. The bait and prey recombinant vectors were transformed into yeast strain *Y2HGold* and *Y187*, respectively. Yeast mating and selection were performed according to the manufacturer’s protocols.

### 4.6. Bimolecular Fluorescence Complementation Analysis (BiFC) Assay

For BiFC, the DNA fragments PaCOL1, PaCOL1ΔB, PaCOL1ΔC, AtCCA1, and AtFLC were introduced into pSAT5-DEST-cEYFP (pE3130) and pSAT4-DEST-nEYFP (pE3136) and pSAT4(A)-DEST-nEYFP (pE3134) and pSAT5(A)-DEST-cEYFP (pE3132) by using the recombination Gateway system (Thermo Fisher Scientific, Waltham, MA, USA). The recombinant constructs in pairs were cobombarded into onion epidermal cells by using a helium biolistic particle-delivery system (model PDS-1000, Bio-Rad, Hercules, CA, USA). Fluorescent signals were examined under an Olympus U-LH 100 HG inverted fluorescence microscope.

### 4.7. Transcriptome Sequencing

Total RNA was extracted from the shoot apex of 40-day-old *Arabidopsis* plants grown under SD conditions with sampling at ZT7 as described [81]. The RNA sequencing library was constructed by using the TruSeq PE Cluster kit (Illumina, Santiago, CA, USA) according to the manufacturer’s instructions. Transcriptome sequencing and assembly involved an Illumina HiSeq 4000 platform (Novogene, Hong Kong, China).

### 4.8. Electrophoresis and Immunoblotting Analysis

The orchid protein extraction buffer contained 2% CTAB, 2% PVP40, 100 mM Tris (pH 8.0), 25 mM EDTA (pH 8.0), 68.8 mM spermidine, 1 mM PMSF, and 1 mM DTT, 1D SDS-PAGE involved using 12.5% (w/v) polyacrylamide gels as described [85]. To generate an antibody against PaCOL1, a DNA construct encoding the 126th to 247th amino acids of PaCOL1 was prepared by PCR and subcloned into the *Bam*HI site of the *pMAL-c5X* expression vector (New England Biolabs, Ipswich, MA, USA) to produce maltose binding protein (MBP) fusion proteins. The MBP fusion proteins were purified according to the manufacturer’s protocols and used to raise a rabbit polyclonal antiserum by LTK BioLaboratories, Taoyuan, Taiwan. The immunoblotting was performed with anti-PaCOL1 antibodies as described [85]. The proteins were quantified by using Bradford assay [86].

### 4.9. Statistical Analysis

Data are shown as mean ± SE from three independent experiments. Statistical differences were analyzed by Duncan’s multiple range test or Student’s *t* test. *P* < 0.05 was considered statistically significant.

## 5. Conclusions

We identified a BBX protein, PaCOL1, as an important regulator for flowering in *Arabidopsis* transgenic plants. PaCOL1 interacted and functioned with CCA1 and LHY in the diurnal rhythmic pathway for flowering. Thus, PaCOL1 plays an important role in integrating signals from photoperiod and stress pathways for flowering in *Arabidopsis*. PaCOL1 may be a good candidate for understanding the signaling pathways for flowering in orchids.

## Figures and Tables

**Figure 1 plants-09-00068-f001:**
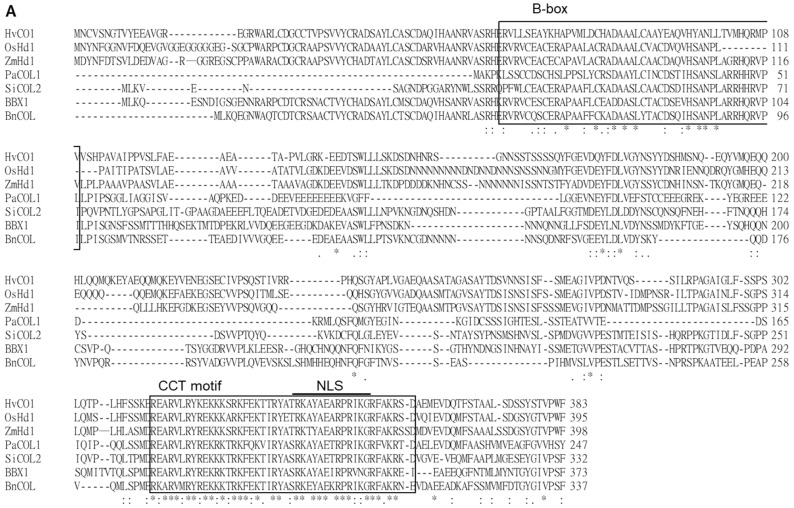
(**A**) Alignment of PaCOL1 and other plant COLs based on deduced amino acid sequences. The multiple sequence alignment of CONSTANS/CONSTANS-like proteins from *Arabidopsis thaliana* (BBX1/AtCO, Q39057), *Brassica napus* (BnCOL, AAP42647.1), *Oryza sativa* (OsBBX18/OsHd1/OsA, Q9FDX8.1), *Sesamum indicum* (SiCOL2, XP_011099077.1), *Hordeum vulgare* (HvCO1, Q8L448), *Zea mays* (ZmHd1, ABW82153.1), and *Phalaenopsis aphrodite* (PaCOL1; ACY95395.1). The consensus amino acids are labeled as ”*” and similar amino acids as “:” or “.” under the sequence. (**B**) Similarity analysis between PaCOL1 and orthologs. (**C**) Phylogenetic analysis of PaCOL1, *Arabidopsis* BBX proteins, and orthologs in other species by using MEGA6. The protein sequences include *A. thaliana* (BBX1-17), *B. napus* (COL and COA1), *Sesamum indicum* L. (SiCOL1 and SiCOL2), *O. sativa* (OsBBX8, OsBBX9, OsBBX10, OsBBX18, OsBBX26, OxBBX27), *Z. mays* (ZmHd1), *Brachypodium distachyon* (BdHd1), *H. vulgare* (HvCO1 and VRN2), *Lolium ternulentum* (LtCO), *Phalaenopsis hybrid* (PhalCOL), and *P. aphrodite* (PaCOL1). Scale bar represents the evolutionary distance. B1 and B2, B-box domain. VP, VP motif. CCT, CCT domain.

**Figure 2 plants-09-00068-f002:**
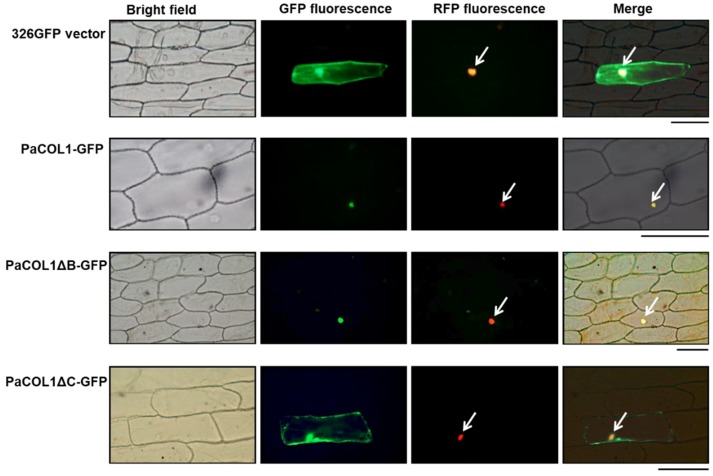
Subcellular localization of PaCOL1 full length, PaCOL1ΔB, and PaCOL1ΔC in onion epidermal cells. The fluorescent signal was examined at 20 h after bombardment. As controls, GFP expression vector and eIF4AIII-RFP fusion protein were used as indicators. Cell nuclei are indicated by an arrow. Scale bar = 200 µm.

**Figure 3 plants-09-00068-f003:**
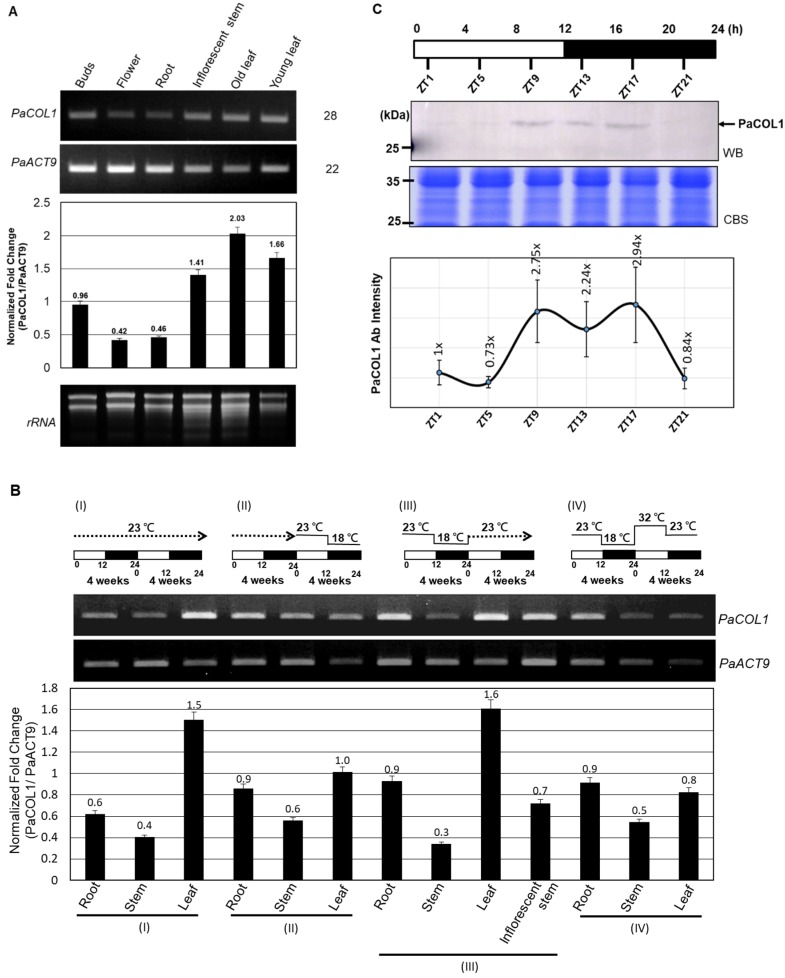
Expression profiles of *PaCOL1* in *P. aphrodite*. (**A**) RT-PCR analysis of *PaCOL1* in tissues of flowering plants grown under 12 h/23 °C light and 12 h/23 °C dark. Top, the image of electrophoresis. Middle, the signal intensity was quantified by ImageJ software. Data are mean ± SE from three independent experiments (three plants in each test). Bottom, rRNA quality control. (**B**) RT-PCR analysis of *PaCOL1* in tissues of mature plants grown under 12 h/23 °C light and 12 h/23 °C dark for four weeks followed by the indicated treatment. Data are mean ± SE expression relative to that of *PaACT9* from three independent experiments. (**C**) Immunoblot analysis of PaCOL1. The protein samples were collected at the indicated time; 120 µg protein was loaded in each lane and immunoblotting involved the PaCOL1 antibody. Zeitgeber time: ZT. Data are mean ± SE expression from three independent biological experiments. WB, Western blot. CBS, Coomassie blue staining.

**Figure 4 plants-09-00068-f004:**
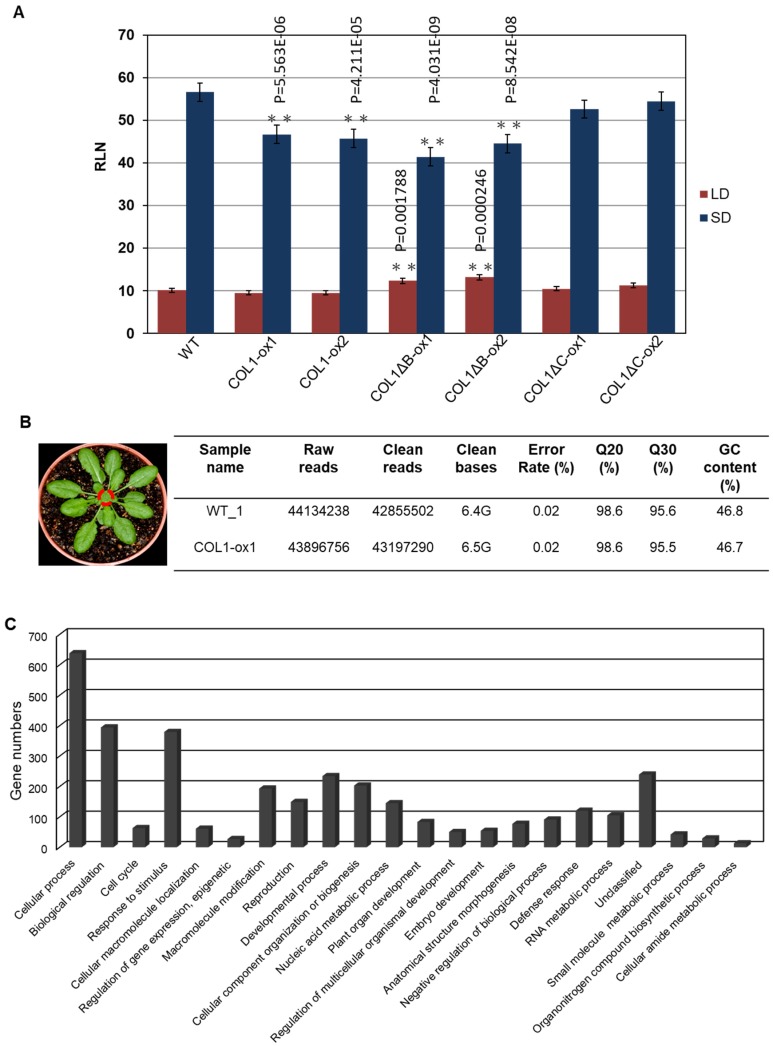
Phenotype analysis of PaCOL1 and two truncated transgenic lines. (**A**) Flowering time of WT, COL1-ox, COL1ΔB-ox, and COL1ΔC-ox plants grown under long-day (LD) and short-day (SD) conditions. Rosette leaf number (RLN) was counted and compared. Data are mean ± SE from three independent experiments (30 plants in each test). * P < 0.05 compared with Col-0 wild type. (**B**) Transcriptome analysis of WT and COL1-ox1 plants. Total RNA was extracted from 40-day-old plants grown under SD for RNAseq analysis. (**C**) Gene Ontology analysis of upregulated genes in 22 functional groups. (**D**) Classification of flowering-related genes in developmental process. (**E**) Real-time PCR analysis of *Arabidopsis AtCCA1*, *AtCO*, *AtSVP*, *AtSOC1*, *AtFLC*, *AtAGL24*, *AtGI*, *AtCDF1*, *AtTFL1*, *AtMAF5*, *AtNF-YB2*, and AtNF-YC3 in 40-day-old WT and COL1-ox1 grown under SD conditions. Data are mean ± SE expression relative to that of AtTUB from three independent experiments (10 plants in each test). ** P < 0.01 compared with the WT. The primers PaCOL1-F and PaCOL1-R were used to check fragments of PaCOL1, PaCOL1∆B, and PaCOL1∆C in *Arabidopsis* transgenic plants and the products were 750 bp, 603 bp, and 630 bp, respectively. The products were identified by sequencing.

**Figure 5 plants-09-00068-f005:**
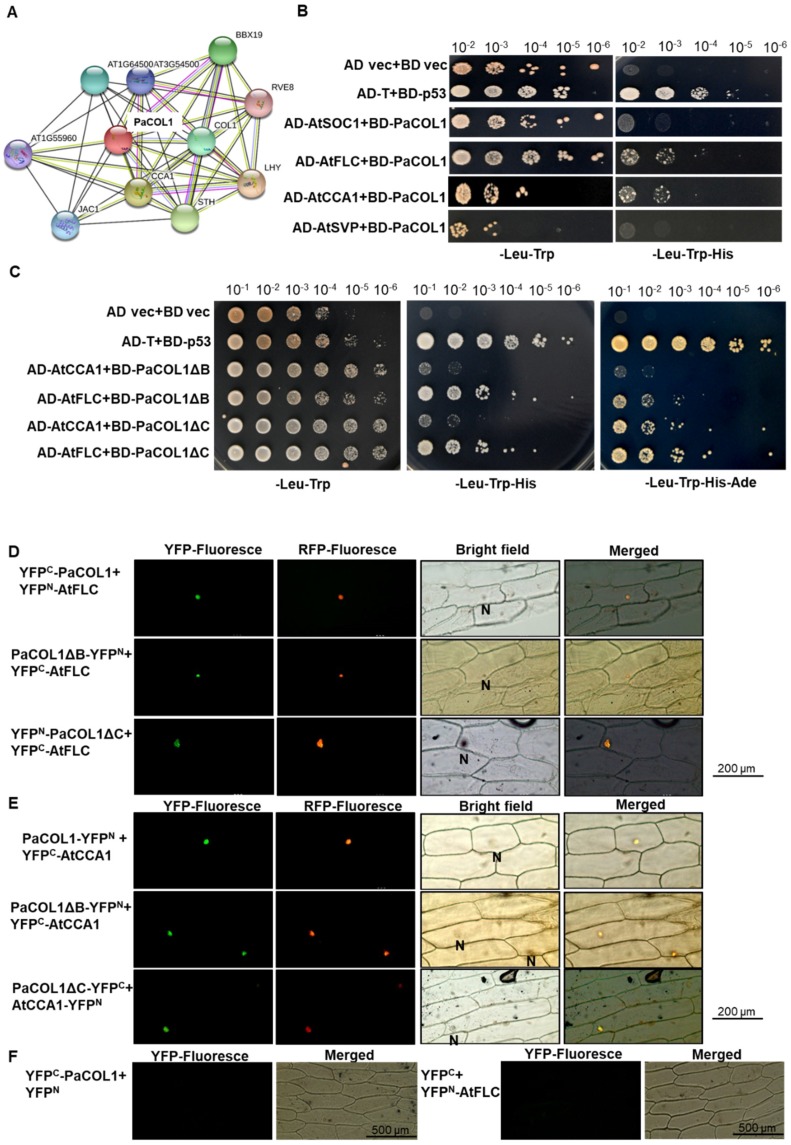
Identification of PaCOL1 and candidate protein interactions. (**A**) Analysis using STRING software for the candidate proteins that interacted with PaCOL1. (**B**) Y2H assay of the interaction between PaCOL1 with AtFLC and AtCCA1. The pGBKT7-53 and pGADT7-T interaction was a positive control. The empty vector interaction was a negative control. (**C**) Y2H assay of the interaction between PaCOL1 truncated proteins (PaCOL1ΔB and PaCOL1ΔC) and AtFLC and AtCCA1, respectively. (**D**) BiFC assay of the interaction between PaCOL1 and truncated proteins (PaCOL1ΔB and PaCOL1ΔC) and AtFLC. (**E**) BiFC assay of the interaction between PaCOL1 and truncated proteins (PaCOL1ΔB and PaCOL1ΔC) and AtCCA1. (**F**) The interactions of YFP^C^-PaCOL1 with YFP^N^ or YFP^C^ with YFP^N^-AtFLC were used for control. eIF4AIII-RFP protein was used as a nucleus indicator. N indicates the nucleus.

**Figure 6 plants-09-00068-f006:**
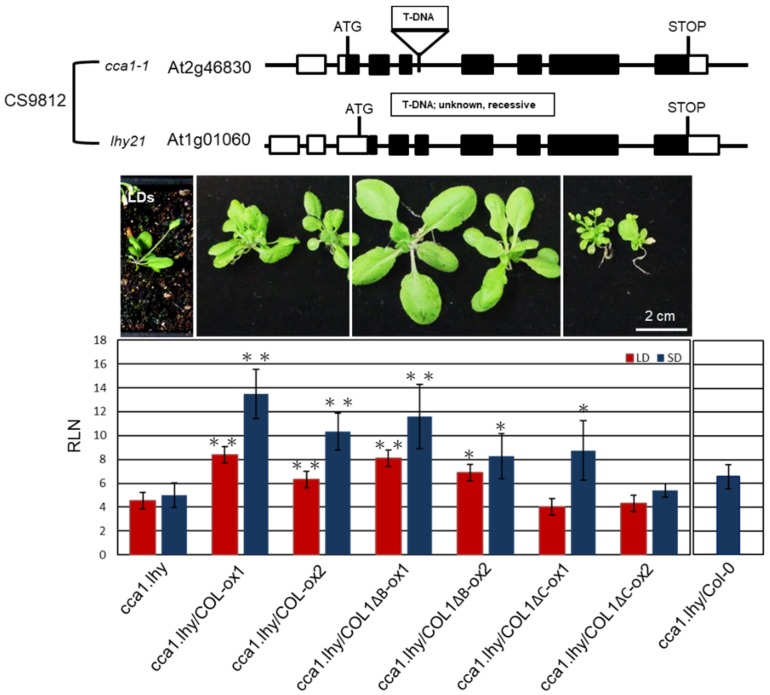
Complementation analysis of *ccal.lhy* double mutant transformed with PaCOL1. Flowering time of *cca1.lhy*, *cca1.lhy*/COL1-ox1, *cca1.lhy*/COL1-ox2, *ccal1lhy*/COL1ΔB-ox1, *cca1.lhy*/COL1ΔB-ox2, *cca1.lhy*/COL1ΔC-ox1, and *cca1.lhy*/COL1ΔC-ox2 plants grown under LD and SD conditions. The *cca1.lhy*/Col-0 hybrid progenies grown under SD conditions were used as control. Rosette leaf numbers (RLN) were counted and compared. Data are mean ± SE from three independent experiments (25 plants in each test). ”**” means P-value <0.01 compared with the *cca1.lhy* double mutant.

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
