# Peer review of "Molecular Characterization and Expression Profile of PaCOL1, a CONSTANS-like Gene in Phalaenopsis Orchid"

_plants, 2020, doi:10.3390/plants9010068_

Round 1
Reviewer 1 Report
In the manuscript by Ke et al., the authors aim to figure out the effect of PaCOL1 gene on many aspects of the flowering in different LD cycles. The overall experiment design was easy to understand but it needs more comprehensive research to lead out the conclusion.
Major:
What is the expression profile of PaCOL1 in the free-running? If we look at the expression profile in constant light (LL) and see a circadian expression, we can clearly see the circadian clock involved in the changes observed in this paper. The control experiment is necessary to conclude the circadian clock involvement in these physiological changes Authors showed only diurnal cycle (LD) here.
Minor:
The truncation of the c-terminal of PaCOL1 (PaCOL1ΔC) may result in unfolding the entire protein if too many residues are truncated. If you truncate the domain little less, sometimes the protein is functional. This is proofreading of the bioinformatics result with a wet experiment. The authors may check it to conclude PaCOL1ΔC is non-functional.
Reviewer 2 Report
The authors successfully presented data that supports that paCOL1 is involved in the flowering pathway.
Here are a couple of general thoughts on the paper.
It is a little confusing why the authors name the protein paCOL1 when the AtCOL1 is of a different BBX group. It suggests that paCOL1 is a homolog AtCOL1, which is rather misleading. If this is the first COL protein identified in Phalaenopsis Aphrodite perhaps it should be clarified early in the manuscript.
There was no comment on possibly knocking out the paCOL1 gene. Was this done and was there any flowering abnormality seen? The overexpression of gene can lead to an imbalance in thresholds that could have many unrelated consequences. The authors did however, repeated qRT-PCR with various overexpressing lines on the selected genes. However, it would be interesting to know if a knockout showed any effects.
Minor comments:
Line 17: … at late afternoon and at the middle of the night.
Figure 2: Some of the outlines of the boxes are different. It should be kept consistent.
Figure 3B: In figure 3A, the inflorescent stem and leaves (old and young) showed a decent amount of paCOL1 expression in the P. Aphrodite. It would be beneficial to show the how that changes in all the different temperature profiles tested and not just in profile III.
Line 304-308: showing this data in the supplementary figures will be beneficial.
Figure 6: The authors should explain in the discussion the differences seen in LD flowering time between the cca1.lhy/COL1ΔC-ox1 and cca1.lhy/COL1ΔC-ox2 lines. This was not mentioned in the paper.
Line 382: I think this statement is over-reaching and not necessary for this paper.
Round 2
Reviewer 1 Report
It is good to publish.